# Pangenome dynamics and population structure of the zoonotic pathogen *Salmonella enterica* serotype Hadar

Kaitlin A. Tagg[1,12], Arancha Peñil-Celis [2,12] ✉, Hattie E. Webb [1] ✉,
G. Sean Stapleton[1], Zachary Ellison [1,3], Molly Leeper[1], Justin Y. Kim[1,4],
Mustafa Simmons[5], Glenn Tillman[5], Cong Li[6], Beth Harris[7],
Brenda R. Morningstar-Shaw[8], Molly K. Steele[1], Daniel Mallal[9],
Shannon Matzinger [9], Kathy Manion[10], John Hergert[11], Jennifer M. Wagner [11],
Colin Schwensohn[1], Joshua M. Brandenburg[1,3], Sheryl Shaw[5],
Katharine Benedict[1], Jason P. Folster[1], Uday Dessai [5],
Santiago Redondo-Salvo [2], M. Pilar Garcillan-Barcia [2] &
Fernando de la Cruz [2]

The bacterial accessory genome, comprised of plasmids, phages, and other mobile elements, underpins the adaptability of bacterial populations. Pangenome (core and accessory) analysis of pathogens can reveal epidemiological relatedness missed by using core-genome methods alone. Employing a *k*-mer-based Jaccard Index approach to compute pangenome relatedness, we explore the population structure and epidemiology of *Salmonella enterica* serotype Hadar (Hadar), an emerging zoonotic pathogen in the United States (U.S.) linked to both commercial and backyard poultry. A total of 3384 U.S. Hadar genomes collected between 1990 and 2023 are analyzed here. Hadar populations underwent substantial shifts between 2019 and 2020 in the U.S., driven by the expansion of a lineage carrying a previously uncommon prophage-like element. Phylogenetic and pangenomic relatedness, coupled with epidemiological data, suggest this lineage emerged from extant populations circulating in commercial poultry, with subsequent dissemination into backyard poultry environments. We demonstrate the utility of pangenomic approaches for mapping vertical and horizontal diversity and informing complex dynamics of zoonotic bacterial pathogens.

The accessory genome, comprising plasmids, prophages, genomic islands, and other mobile genetic elements (MGE), is a key component of bacterial evolution[1]. While typically excluded from phylogenetic or source attribution analyses[2,3], there is growing interest in the discriminatory and predictive power of the accessory genome for epidemiological investigations[4–7]. For zoonotic pathogens like *Salmonella enterica* with numerous transmission routes[8–10], analysis of the pangenome (accessory and core genome) has proven useful for

enhanced surveillance, outbreak investigation, and microevolutionary exploration[11–13]. The added public health value of pangenome data, however, depends on the unique genomic structure and microbial ecology of each *Salmonella* serotype and should be assessed within the context of serotype-specific population analyses. High-resolution pangenomic analyses, coupled with epidemiological and source information, are likely to be particularly informative for serotypes linked to multiple sources and transmission pathways or

for clonal lineages that exhibit limited variability in their core genome[4], such as *S. enterica* serotype Hadar (herein referred to as Hadar).

Hadar is transmitted to people via contaminated food and contact with animals and has caused several United States (U.S.) outbreaks in the last decade, linked to either ground turkey consumption or contact with backyard poultry (i.e., privately-owned, non-commercial poultry such as chickens, ducks, or turkeys)[14,15]. Although Hadar is considered a highly clonal serotype, exhibiting limited variability by core-genome multilocus sequence typing (cgMLST)[14], strains transmitted by these two different sources were historically differentiable (allele range 25-50). However, in 2020, despite decreased reporting of enteric illness during the early years of the coronavirus disease 2019 (COVID-19) pandemic, an emergent Hadar strain was linked with both ground turkey consumption and backyard poultry contact. These outbreaks resulted in >900 human illnesses compared to <500 total reported cases of Hadar in all years prior to 2020[14,16]. Traceback investigations were not able to determine the epidemiological connection suggested by the detection of indistinguishable strains (determined by cgMLST) from two ostensibly distinct sources: commercial poultry and backyard poultry[14,15]. This emergent strain, now responsible for >2000 human illnesses, continues to cause outbreaks into 2024; it has been designated by the U.S. Centers for Disease Control and Prevention (CDC) as a Reoccurring, Emerging, or Persisting (REP) strain REPTDK01, with a cgMLST range of 0–26 allele differences[17].

In this work, given the limitations in discriminatory power of cgMLST for this strain, we employ *k*-mer-based Jaccard Index (JI) to compute pangenome relatedness[13] of Hadar along the U.S. farm-to-fork continuum (Farm-to-Fork Continuum). We explore and assess the value of the pangenome for delineating strains, for attributing human cases to transmission vehicles, and for a general understanding of the epidemiological and microevolutionary dynamics that underpin Hadar disease incidence and environmental persistence. In addition, we build a foundational landscape of the vertical and horizontal diversity and dynamics of this serotype and offer support for the incorporation of the accessory genome for differentiating strains transmitted via different pathways.

## Results

### Pangenome structure of United States Hadar population

Hadar genomes self-organized into 18 clusters by JI (JI threshold=0.988), labeled JI-A through R (Fig. 1); less than 5% of genomes (*n* = 158/3387) did not cluster with a JI-group and were considered singletons (Fig. 1). To better understand the relationships among these clusters, the three largest groups JI-A, JI-B, and JI-C were further divided into subgroups using an increased JI threshold (Supplementary Fig. 1). JI-A subgroups A1-15 were defined at JI = 0.995; JI-B subgroups B1-6 and JI-C subgroups C1-9 were defined at JI = 0.992. The MGE that define each JI-group include large plasmids (>30 kb), prophages, integrative conjugative elements (ICE), antimicrobial resistance (AMR) regions, or regions of unknown function (Fig. 2). In some cases, two JI-groups differed only by the presence of a large plasmid (e.g., JI-A and JI-C; JI-B and JI-G; JI-D and JI-E), while others displayed more differences in their pangenome content (e.g., JI-I) (Fig. 2).

To contextualize the pangenome with core lineage information, ST (sequence type, based on 7 core loci), National Center for Biotechnology Information (NCBI) SNP (single nucleotide polymorphism) cluster[18], and cgMLST allele code (based on *n* = 3002 core loci)[3] were separately visualized on the network. Over 98% of Hadar genomes in this analysis are ST33 (*n* = 3326/3384); only JI-I (ST473), JI-L (ST5130 and ST9222), and JI-Q (ST473) contained genomes of a different ST (Supplementary Fig. 2a). NCBI SNP cluster aligned well with JI-groups; PDS000158107 was the most common cluster, encompassing the largest groups JI-A, JI-B and JI-C (Fig. 3a). cgMLST allele codes also aligned well with JI-groups, with the majority of groups (*n* = 12/18) containing a single condensed allele code (Fig. 3b). Despite being in the same NCBI SNP cluster, JI-A and JI-C separate from JI-B by condensed allele code (Figs. 3a and 3b). Both NCBI SNP cluster and cgMLST suggest membership within certain JI-groups is due to convergence in pangenome content rather than core genome similarity.

Furthermore, plasmids were common in U.S. Hadar genomes, with 60% (*n* = 2047/3384) containing one or more Col-like plasmids and 22% (*n* = 740/3387) carrying at least one large (>30 kb) conjugative plasmid (Fig. 3c, and Supplementary Fig. 2b). IncI1 was the most common replicon, detected in two different Plasmid Taxonomic Units (PTUs): PTU-I1, present in JI-C, JI-E, and JI-G, and an unnamed PTU

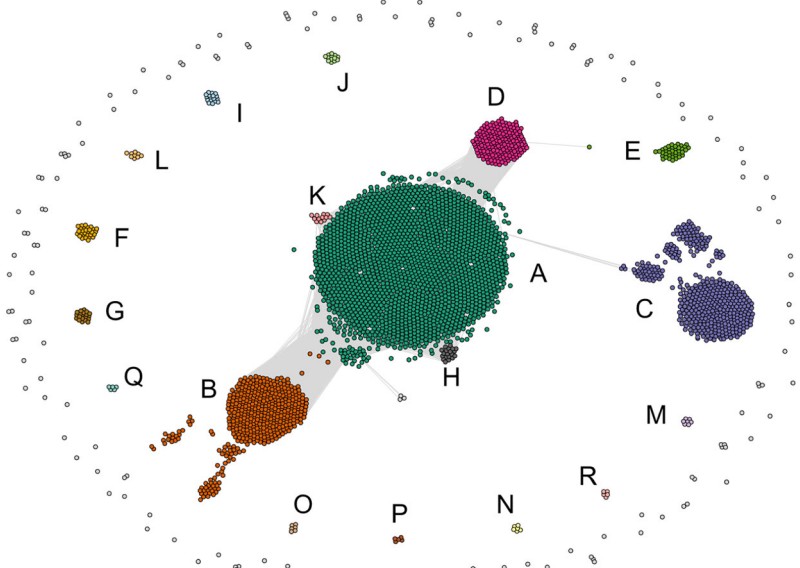

| JI-group | Count (%) | SNP cluster | PTU |
|---|---|---|---|
| A | 1899 (56.1) | PDS000158107 [1] | none |
| B | 489 (14.5) | PDS000158107 [1] | none |
| C | 453 (13.4) | PDS000158107 | PTU-I1 |
| D | 191 (5.6) | PDS000114062 | PTU-X1 |
| E | 40 (1.2) | PDS000114062 | PTU-I1, PTU-X1 |
| F | 29 (0.9) | PDS000014162 | none |
| G | 20 (0.6) | PDS000158107 | PTU-I1 |
| H | 20 (0.6) | PDS000158106 | none |
| I | 17 (0.5) | PDS000001789 | PTU-E78, PTU-NA |
| J | 13 (0.4) | PDS000014162 | PTU-NA |
| K | 12 (0.4) | PDS000158107 | none |
| L | 9 (0.3) | PDS000043817 | none |
| M | 7 (0.2) | PDS000158106 | none |
| N | 6 (0.2) | PDS000158107 | PTU-NA |
| O | 6 (0.2) | PDS000158106 | none |
| P | 5 (0.1) | PDS000129867 | none |
| Q | 5 (0.1) | PDS000023881 | none |
| R | 5 (0.1) | PDS000158107 | none |
| singleton | 158 (4.7) | none | none |
| **TOTAL** | **3384 (100)** | **--** | **--** |

[1] Dominant SNP cluster (>90% of genomes)

**Fig. 1 | Distribution of *Salmonella* Hadar genomes by Jaccard Index (JI).** The network contains 3384 nodes, connected when JI ≥ 0.988. Eighteen groups (JI-groups A-R) are labeled, singleton genomes that do not associate with a JI-group are displayed around the outside of the network. Genomes are colored according to JI-group. Counts and percentages of genomes within each JI-group, along with their dominant National Center for Biotechnology (NCBI) SNP (single nucleotide polymorphism) cluster and plasmid taxonomic unit (PTU) profile are included in the table.

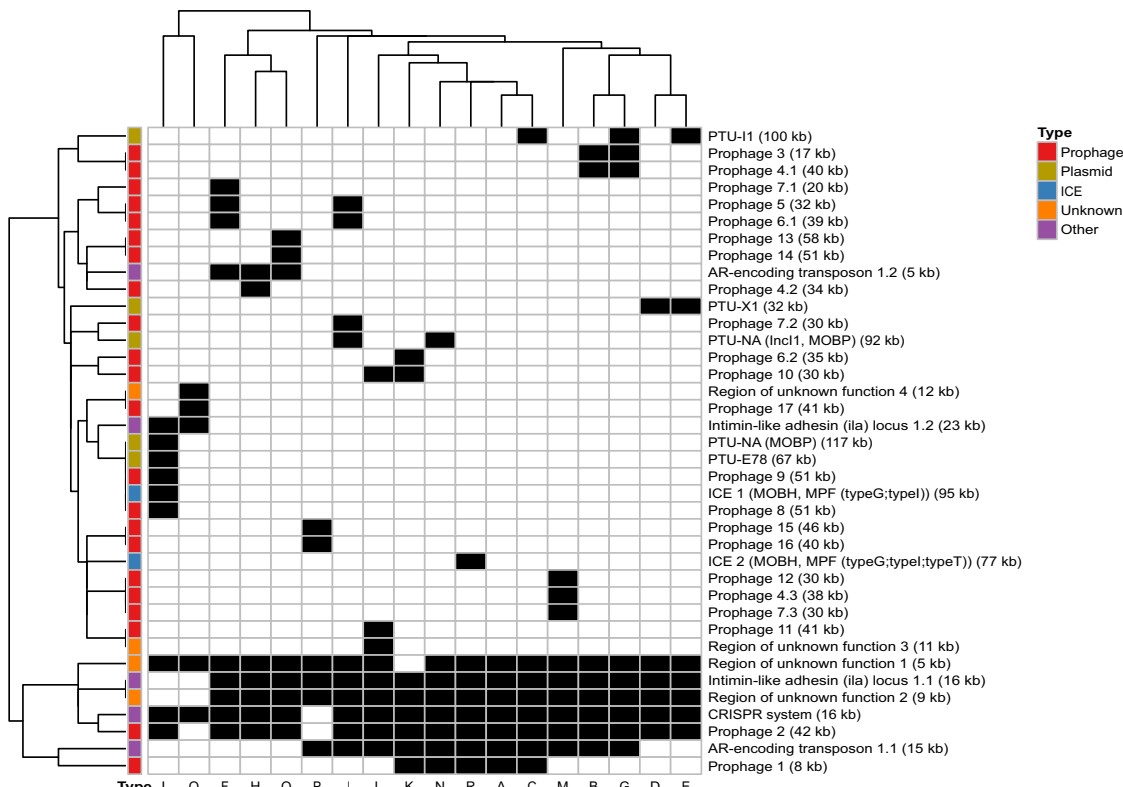

**Fig. 2 | Differential distribution of accessory genome elements in Jaccard Index (JI)-groups.** The heatmap displays the presence or absence of indels detected using PanGraph[25]. Indels larger than 5 kilobase (and their variants) were included in the analysis. Each column represents a JI-group (labeled along the bottom axis). Each row corresponds to an indel; presence in the corresponding JI-group is indicated in black, and absence indicated in white. The left bar categorizes the indels as "plasmid", "prophage", "ICE" [Integrative and Conjugative Elements], "other", or "unknown", as represented in the legend. Variants of named indels are indicated with a digit (e.g., Prophage 7.2).

(PTU-NA: IncI1, MOB$_P$) found in JI-J and JI-N (Fig. 2, Fig. 3c). JI-I also contained PTU-E78, a recently identified non-mobilizable PTU, and another unnamed PTU (PTU-NA: MOB$_P$) (Figs. 2, 3c). Despite the prevalence of plasmids, nearly 30% of genomes ($n = 1011/3384$) contained neither plasmid replicons nor MOB relaxase genes (Supplementary Data 1); these genomes predominantly fell into JI-A (Supplementary Fig. 2b).

In terms of AMR, over 90% ($n = 3055/3384$) of genomes contained at least one AMR determinant; predicted resistance to aminoglycosides (specifically, streptomycin) and tetracyclines was the most common profile, mediated by *aph(3″)-Ib*, *aph(6)-Id*, and *tet*(A), all integrated in the chromosome (Fig. 3d, and Supplementary Data 1). Predicted resistance to penicillins was less common (4%, $n = 127/3384$) and was predominantly mediated by $bla_{TEM-1}$ (Supplementary Fig. 2c and d). While rare, cephalosporin resistance mediated by $bla_{CMY-2}$ was detected in groups JI-C and JI-E (0.4%, $n = 12/3384$; Supplementary Fig. 2c and d). Members of JI-D, JI-I and JI-Q were predicted to be pansusceptible, with no known AMR determinants detected (Supplementary Data 1).

### Genetic and epidemiological differences between pangenome groups

The dominant pangenome groups changed substantially between 2016 and 2023, most notably between 2019 and 2020 (Fig. 4, Fig. 5, and Supplementary Fig. 2e and f). This shift was particularly pronounced for human and retail meat samples, where JI-A and JI-C were rare prior to 2020 yet comprise between 56% and 100% of samples collected in years 2020-2023 (Fig. 4). JI-B was the most common group detected in retail meat and animal (cecal) sampling prior to 2020 but decreased in detection substantially in 2020-2023; JI-B was not detected at all in

2023 retail meat sampling. Groups JI-D and JI-E made up more than half of human Hadar samples in 2016 and 2017 but have not been detected since 2019; these groups were not found in retail meat or animal sampling throughout the study years (Fig. 4). JI-A and JI-C are the most common JI-groups in all three sampling systems from 2020-2023. This pattern is clear across the country, where JI-A was geographically restricted and relatively rare prior to 2020, then underwent nationwide expansion to become the dominant group in nearly all states by 2020-2023 (Fig. 5). Conversely, JI-B contracted from a widespread geographic distribution to more limited regional presence, with complete absence from several states where it was previously detected.

JI-A and JI-C are indistinguishable by cgMLST-based phylogeny (Fig. 6: Ring 1) but differ in their pangenome by carriage of a ~100 kb PTU-I1 (IncI1) plasmid, which underpins the separation of these two JI-groups (Figs. 2 and 3c). Most JI-A and JI-C genomes fall within a comparatively tight "emergent" clade that forms the CDC-defined REPTDK01 strain (Fig. 3e, Fig. 6), associated with ground turkey consumption and backyard poultry contact based on previous multistate outbreak investigations[17]. Of interest, two temporally "ancestral" JI-A genomes isolated from wild ducks in 1990 are positioned in a clade adjacent to the emergent genomes (Fig. 6). This emergent clade invariably contains an ~8 kb prophage, labeled here prophage 1 (Supplementary Data 4), that forms part of the core pangenome of JI-A and JI-C (Fig. 2). Prophage 1 was detected as early as 2004 in singleton Hadar genomes (imported "sweet good without custard or cream filling" from Pakistan), was seen in genomes from swine and commercial poultry samples from 2015, yet remained uncommon until the 2020 emergence of REPTDK01 (Fig. 6, and Supplementary Fig. 3). According to PHASTEST, prophage 1 is related to filamentous

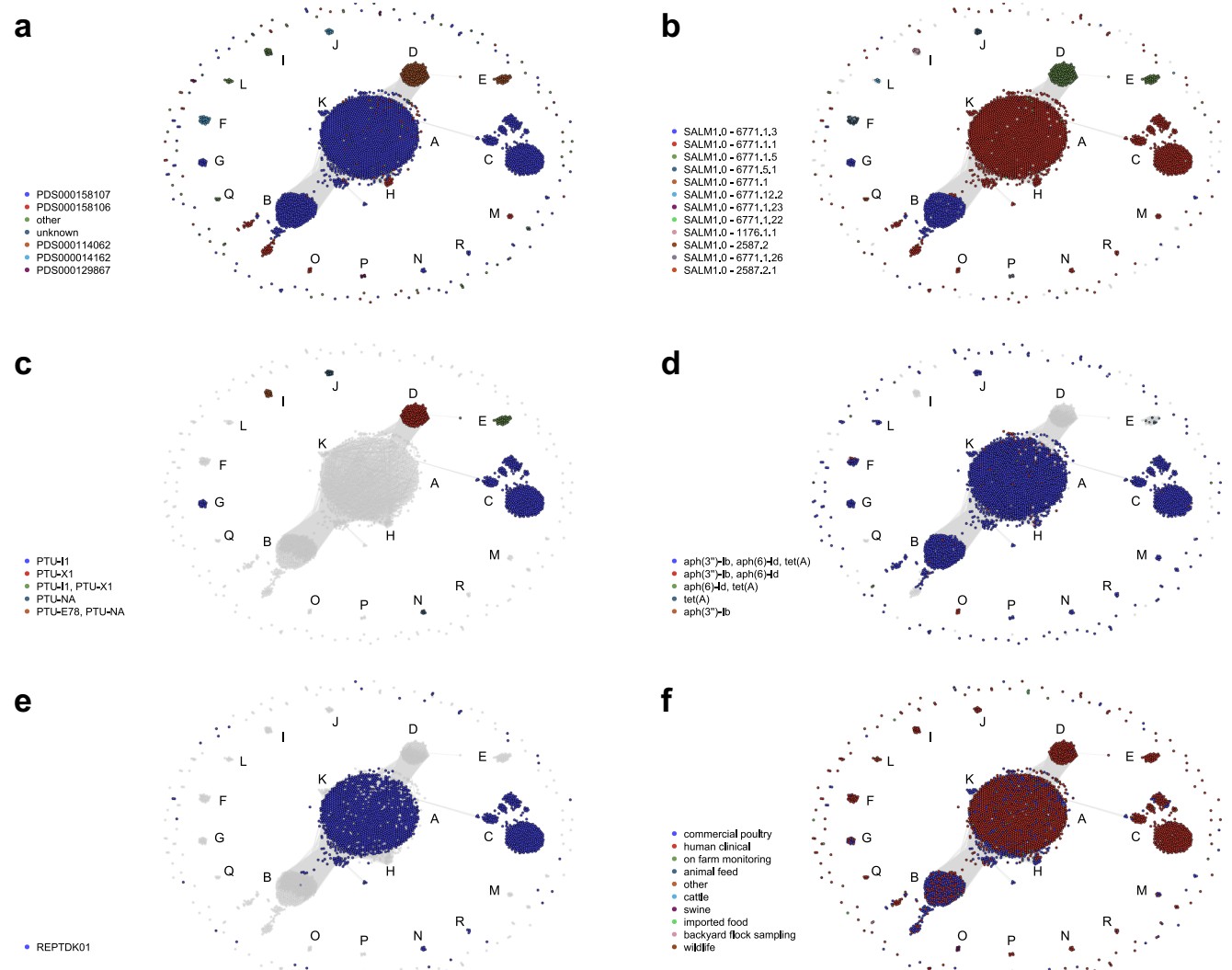

**Fig. 3 | Distribution of Hadar genomes by variables of interest.** The networks contain 3384 nodes, connected when Jaccard Index (JI) ≥ 0.988. Eighteen groups (JI-groups A-R) are labeled. Legends are left of each network. **a** National Center for Biotechnology Information (NCBI) SNP (single-nucleotide polymorphism) cluster across JI-groups (obtained from Pathogen Detection Isolate Browser[45]). **b** Condensed allele code across JI-groups. Grey nodes indicate genomes with no assigned allele code. **c** Plasmid taxonomic units (PTUs) across JI-groups. Grey nodes indicate genomes with no known PTU. **d** Most common antimicrobial resistance (AMR) genes conferring predicted aminoglycoside and tetracycline resistance. Grey nodes indicate genomes without selected AMR genes. **e** Genomes determined to be REPTDK01 strains according to Centers for Disease Control and Prevention (CDC)-defined core genome multilocus sequencing typing (cgMLST) allele range. Grey nodes indicate genomes not assigned to REPTDK01. **f** Genomes isolated from different sources.

phages I2-2 and Ike, and contains a protein with N-terminal homology to the zonular occludens toxin protein (Zot) (Supplementary Fig. 4). The phage-encoded Zot proteins in *Vibrio cholerae*[19] and *Campylobacter* spp.[20,21] have a demonstrated pathogenic role attributable to a C-terminal enterotoxic domain[22]. While homology with Zot proteins does not imply toxigenic function, the Hadar Zot-like protein identified here was bioinformatically predicted to contain toxigenic regions using ToxinPred3.0[23], hinting at a putative role in pathogenesis. Thus, prophage 1 presence is notable both from an epidemiological and biological perspective, and its pathogenic and adaptive capacity is being assessed with functional analysis.

JI-B is the second most abundant pangenome group, predominantly encompassing genomes from commercial poultry (Fig. 1, Fig. 3f). A smaller group, JI-G, is indistinguishable from JI-B phylogenetically (Fig. 6: Ring 1) but can be differentiated by the presence of PTU-I1 (IncI1) plasmids (Fig. 2). JI-B (and JI-G) genomes appear more diverse in their core genome relative to those from other dominant pangenome groups (e.g., JI-A, JI-C, JI-D and JI-E) (Fig. 6), which may be a reflection of time and environmental factors−

genomes in JI-B were isolated as early as 2011 from poultry sources across the country (Supplementary Data 1). Analysis of JI-B subgroups did not reveal any geographic association (Supplementary Fig. 1b) or link to specific processing facilities. Of note, genomes from human samples that were part of a 2019 multistate Hadar outbreak linked to ground turkey consumption (internal CDC investigation) all fell into JI-B or JI-G, suggesting Hadar strains from these groups are transmitted via food.

In contrast, groups JI-D and JI-E were almost exclusively from ill humans (n = 6/191 JI-D genomes are from non-human sources) (Fig. 3f), with upwards of 40% of cases reporting contact with backyard poultry (*n* = 79/191 JI-D genomes; n = 24/40 JI-E genomes, Supplementary Data 1). JI-D and JI-E genomes display relatively little core diversity (Fig. 6) and differ from each other only by the carriage of PTU-I1 (IncI1) plasmids (Fig. 2). They differ from other JI-groups phylogenetically in that they are encompassed in a single clade by core SNP analysis (Fig. 6); and they differ pangenomically in that they lack a common AMR region ("AMR-encoding transposon [Tn] 1.1", Fig. 2) and are the only groups to carry PTU-X1 (IncX1) plasmids (Fig. 2). Genomes in these

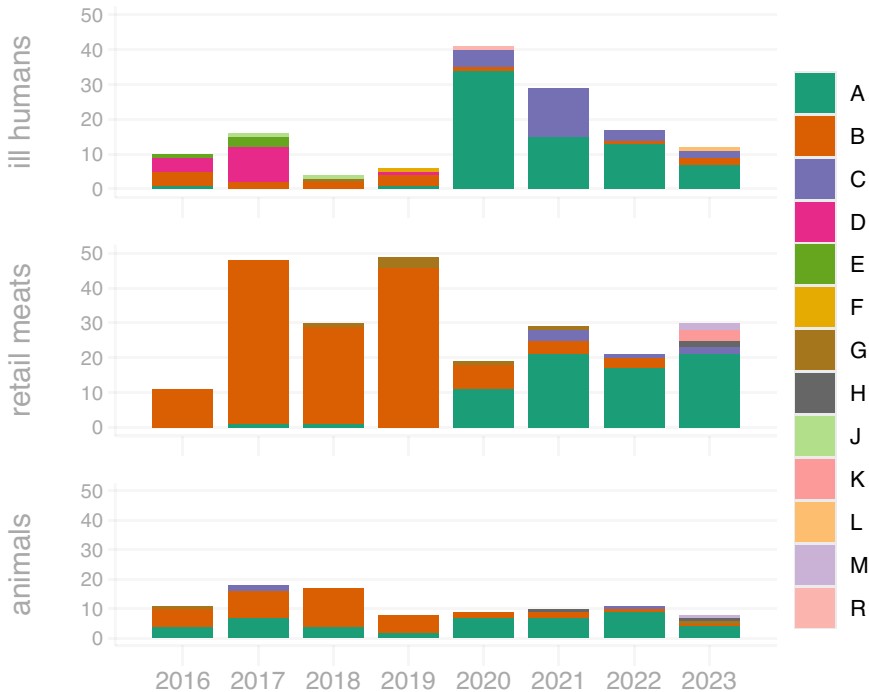

**Fig. 4 | Abundance of Jaccard Index (JI)-groups over time.** National Antimicrobial Resistance Monitoring System (NARMS) data from Centers for Disease Control and Prevention (humans), U.S. Food and Drug Administration (retail meats) and U.S. Department of Agriculture's Food Safety and Inspection Service (animals) from 2016–2023 are included. Years are displayed on the x-axis and counts of isolates according to JI-group are displayed on the y-axis. Source data are provided as a Source Data file.

groups were part of 2016 and 2017 multistate outbreaks linked to contact with backyard poultry[24].

Two small pangenome groups, JI-H and JI-K, are of interest because of their connectivity to JI-A in the network, indicating pangenomic relatedness (Fig. 1). JI-H genomes are all from commercial chicken sampling or from ill humans (no exposure information available), representing a statistically significant "chicken-source cluster" (Supplementary Data 1; p < 0.00001, chi-squared) that is unique among the more common commercial turkey source. JI-K genomes were all isolated throughout 2023, are almost exclusively from turkey product samples ($n = 11/12$) and are predominantly from a single state ($n = 8/12$ were isolated in California) (Fig. 5, Supplementary Data 1). JI-K genomes carry prophage 1, along with two other larger prophages unique to this group (prophage 6.2 and prophage 10; Fig. 2), potentially representing recent divergence from REPTDK01.

Several pangenome groups harbor PTU-I1 (IncI1) plasmids, including JI-C, JI-E and JI-G (Fig. 1, Fig. 2). PTU-I1 (IncI1) plasmids are common in avian environments, often carry AMR genes, and may play a role in virulence and growth inhibition of competing bacteria[25,26]; thus, their presence and diversity in this dataset were of interest. Core plasmid SNP analysis coupled with AcCNET (Accessory Genome Constellation Network) plasmid proteome analysis were used to assess the relatedness of PTU-I1 (IncI1) plasmids between and within JI-groups (Fig. 7). PTU-I1 (IncI1) plasmids from all three JI-groups were surprisingly diverse in their core and proteome. They did not form phylogenetic clades defined by the JI-group of the host, instead plasmids from the same JI-group were clustered in different clades (Fig. 7a). Notably, nearly identical plasmids were found in isolates recovered from different environments; for example, in a JI-G isolate from commercial poultry (FSIS11705123) and in a JI-E isolate from a human clinical case with reported backyard poultry contact (PNU-SAS013855). PTU-I1 (IncI1) plasmids also intermingled phylogenetically with those from other Enterobacteriaceae species (Fig. 7,

Supplementary Data 5); for instance, one *E. coli* plasmid was > 99.9% identical to a JI-C1 plasmid (Fig. 7a). These findings support the notion that PTU-I1 (IncI1) plasmids move horizontally between strains circulating in different environments and across different bacterial species. In contrast to this diversity observed across JI-groups and species, plasmids from the same JI-C subgroups clustered together phylogenetically (Fig. 7a and Supplementary Fig. 5a) and proteomically (Fig. 7c), indicating that plasmid content is responsible for JI-C subgrouping. Of note, the largest JI-C subgroup, JI-C1, likely represents a multiyear clonal expansion event, given the tight relatedness of its plasmids and chromosomal genome (Fig. 7a and Supplementary Fig. 5).

**Increased discriminatory power for public health investigations**
REPTDK01 was clearly detectable in the pangenome network—98% ($n = 2148/2194$) of these genomes fell into JI-A, JI-C, JI-N and JI-R (Fig. 3e, Supplementary Data 1)—genetically corroborating and adding confidence to the REPTDK01 definition using pangenomic data. Additionally, REPTDK01 was further stratified by JI-grouping and JI-subgrouping, revealing clear epidemiological patterns. For example, while JI-A itself was not statistically associated with either commercial or backyard poultry (Supplementary Table 2), JI-A2 contained predominantly commercial poultry-related genomes from the U.S. and Canada ($n = 42/68$), and none of the human clinical cases in this group ($n = 24/68$) reported backyard poultry contact. In contrast, JI-A3 was almost exclusively comprised of genomes from human clinical cases ($n = 27/28$), a third of which reported backyard poultry contact, and zero commercial poultry-related genomes fell into this group (Supplementary Data 1 and Fig. S1a). JI-N genomes were all human clinical—mostly isolated from the northeast (Fig. 5b; $n = 4/6$)—and may represent a closely related subcluster of illnesses that differ from JI-A REPTDK01 strains only by the carriage of a large plasmid (PTU-NA, IncI1) (Fig. 2). JI-C was significantly associated with backyard poultry ($p < 0.00001$, chi-squared; Supplementary Table 2), representing a

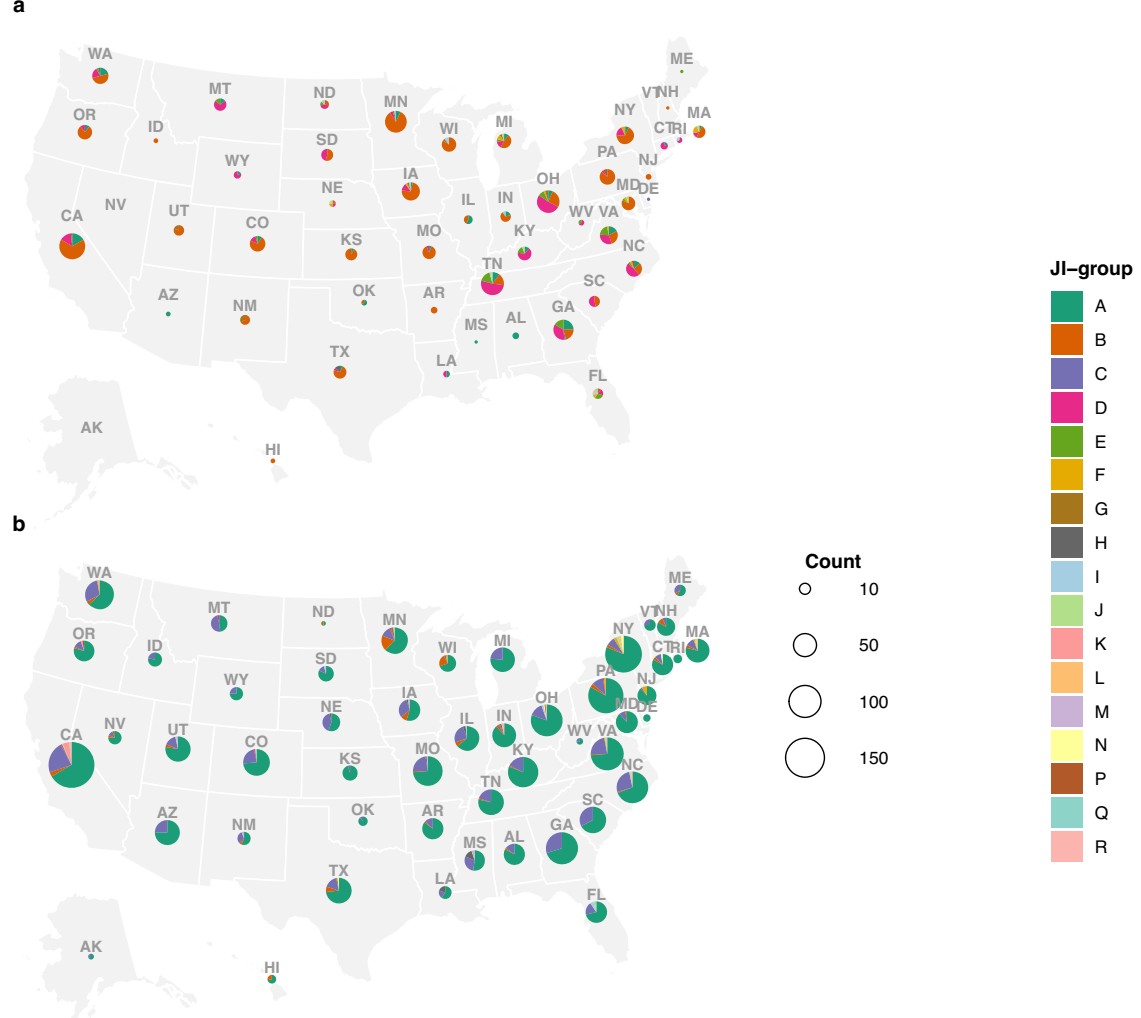

**Fig. 5 | Genomic distribution of *Salmonella* Hadar Jaccard Index (JI)-groups by state. a** Years 2016–2019 (*n* = 701). **b** Years 2020–2023 (*n* = 2342). Pie chart colors show the proportional representation of genomic clusters (JI-groups A-R) within each state differentiated by geographic location and state abbreviation, with chart size proportional to total genome count (range:0-192 genomes per state). Notable geographic shifts include JI-A expansion from limited to nationwide distribution and corresponding JI-B contraction, illustrating the rapid spread of emergent REPTDK01-associated lineages. Data include genomes from national surveillance systems: Centers for Disease Control and Prevention (CDC) PulseNet USA, CDC and U.S. Food and Drug Administration (FDA) National Antimicrobial Resistance Monitoring System (NARMS), and U.S. Department of Agriculture's Food Safety and Inspection Service. Data from Washington D.C. and Puerto Rico are not shown.

subgroup of REPTDK01 (defined by the carriage of PTU-I1 (IncI1) plasmids) that was likely transmitted to humans via animal contact rather than food. More specifically, epidemiological traceback data available for clonal subgroup JI-C1 did not identify a single backyard poultry supply store chain or hatchery, instead suggesting a common reservoir of Hadar upstream of hatcheries. Coupling pangenome data and epidemiological data, REPTDK01 strains can be further differentiated for both retrospective and prospective investigations.

Several other non-REPTDK01 pangenome groups are statistically associated with a specific source or exposure. JI-B and JI-G are each significantly associated with commercial turkey (*p* < 0.00001, chi-squared); JI-B genomes had 17.5 times (95% confidence intervals (CI): 13.7-22.3), and JI-G genomes had 5.9 times (95% CI: 2.1-17.1) higher odds of being from commercial turkey compared with all other JI-groups (Supplementary Table 2). Coupled with the absence of human cases reporting backyard poultry contact in these groups, it is likely that Hadar strains from JI-B and JI-G are acquired through foodborne transmission. In contrast, JI-D and JI-E were each significantly associated with backyard poultry contact (*p* < 0.00001, chi-squared); JI-D

genomes had 2.6 times (95% CI: 1.9-3.6) and JI-E genomes had 5.2 times (95% CI: 2.7-10.6) greater odds of backyard poultry contact, relative to all other JI-groups (Supplementary Table 2). The stark lack of genomes from commercial poultry sources (only JI-D had a single commercial chicken source genome), and the predominance of backyard poultry-associated outbreak genomes in these groups (*n* = 140/191 in JI-D, *n* = 35/40 in JI-E), strongly suggests JI-D and JI-E strains of Hadar are transmitted through animal contact. It is important to note that cgMLST differentiates JI-B and JI-G genomes from JI-D and JI-E (Fig. 3a). Thus, the pangenome analysis performed here provides additional genomic confidence in these attributions.

A handful of small JI-groups contained genomes from humans with limited epidemiological information, but with one or two genomes from a known source. Specifically, both JI-F and JI-J contain a genome from raw dog food (containing duck) obtained from ad hoc pet food sampling (see "Methods") (Supplementary Data 1). JI-L contains two genomes from imported shrimp (Ecuador) isolated in 2022 (Supplementary Data 1). Given the close relatedness of genomes within JI-groups (median

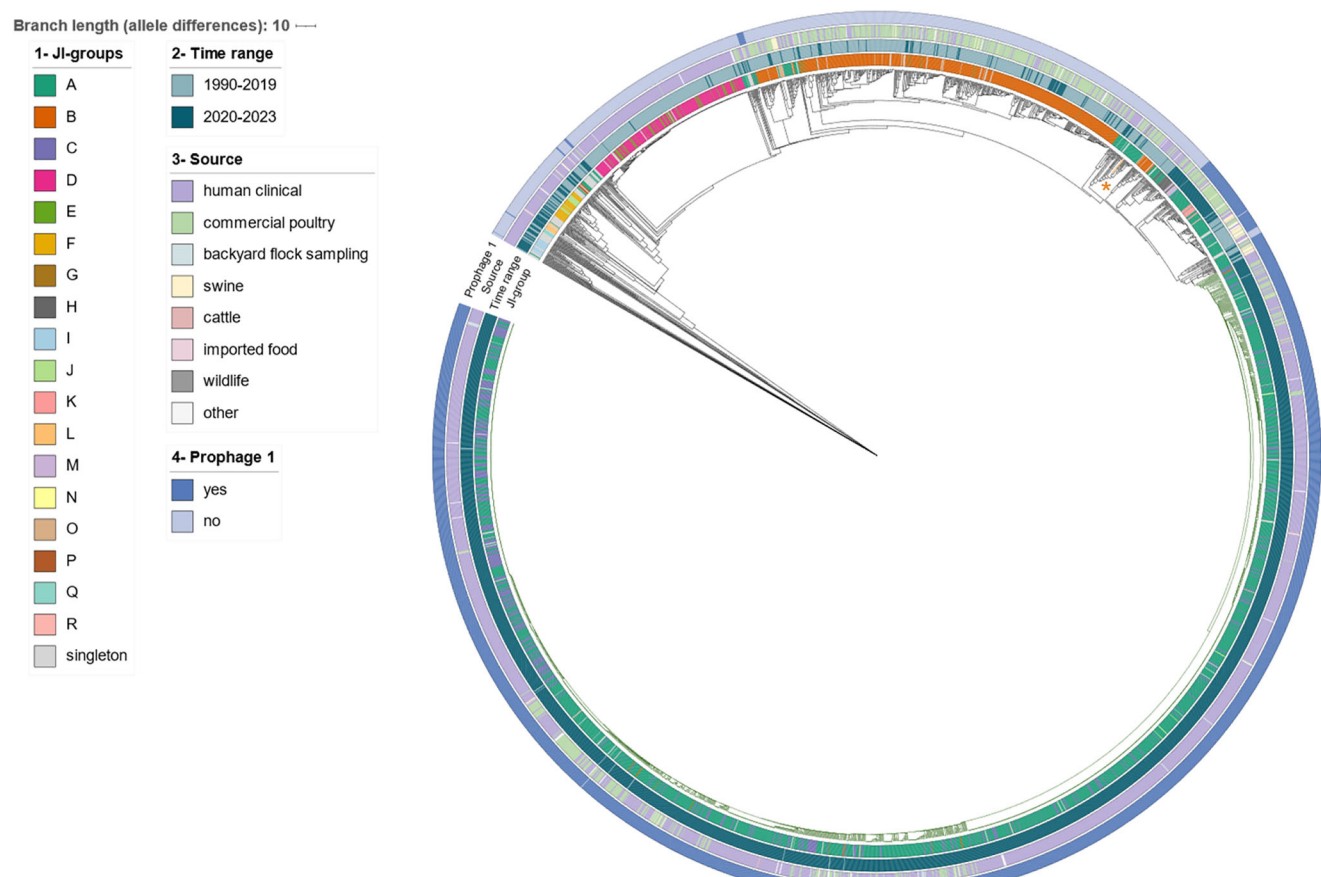

**Fig. 6 | Core genome multilocus sequencing typing (cgMLST)-based phylogenetic tree.** 3363 Hadar genomes are included in the tree (21/3384 could not be processed through BioNumerics (Supplementary Data 1)): Ring 1 displays Jaccard Index (JI)-group, Ring 2 displays time range (1990–2019 versus 2020–2023), Ring 3 displays source, and Ring 4 displays presence of prophage 1 detected in this study. The large clade colored in green represents REPTDK01. Two "ancestral" genomes collected in 1990 are shown with orange branches and an asterisk at -1 o'clock. A list of genomes included in this tree is available in Supplementary Data 1. Tree was generated using BioNumerics v7.6.3 and visualized in iTol v6[42].

average nucleotide identity within JI-groups is ≥ 99.95, Supplementary Fig. 6), the presence of the pet food and imported food genomes alongside genomes from human samples is suggestive of an epidemiological connection, though without exposure information reported by these ill people this link cannot be confirmed. Prospectively, the relatedness of additional human cases found to be within the JI-F and JI-J groups could inform which food items to assess during supplementary interviews of ill people included in an outbreak investigation.

As mentioned above, several pairs of JI-groups differ only by the presence of PTU-I1 (IncI1) plasmids: JI-A and JI-C (plasmid present), JI-D and JI-E (plasmid present), JI-B and JI-G (plasmid present). We further assessed these pairs for epidemiological patterns associated with plasmid presence, including source of isolation, geographic region, and patient demographics where applicable (age, sex, hospitalization), but no variables were significantly different between paired groups ($V < 0.3$, corrected Cramer's $V$; $p > 0.005$, chi-squared). However, PTU-I1 (IncI1) plasmids were independently associated with backyard poultry-related sources (PTU-I1 $n = 208$, no PTU-I1 $n = 526$) when compared with commercial poultry sources (PTU-I1 $n = 32$, no PTU-I1 $n = 699$), and when compared with all other sources (PTU-I1 $n = 305$, no PTU-I1 $n = 2345$) (Supplementary Data 1; $p < 0.00001$, chi-squared). Thus, PTU-I1 (IncI1) plasmids have statistical support to serve as a genetic marker to distinguish strains transmitted via backyard poultry contact versus those more likely

attributed to another source, which is of particular value for differentiating REPTDK01 strains that can be transmitted via several pathways.

## U.S. Hadar pangenome structure reflects a subset of global diversity

A dataset of Hadar genomes ($n = 1145$) from 33 countries other than the U.S., isolated from 1950 through 2023, was used to assess differences in pangenome structure between separate geographical locations (Supplementary Fig. 7, Supplementary Data 2). The non-U.S. dataset partially overlapped with U.S. genomes: 33% of non-U.S. genomes clustered within JI-groups identified in the U.S. pangenome data, while 47% formed distinct JI-groups not present in the U.S. dataset (Supplementary Fig. 8, Supplementary Table 3). The non-U.S. dataset contained 3095 genes absent from the U.S. pangenome, while the U.S. dataset contained 1628 genes absent from the non-U.S. dataset (Supplementary Fig. 9). Both datasets exhibited moderately open pangenomes (Heaps' law γ value ~ 0.2) and shared a core of 4187 genes. Notably, separate analysis of each dataset revealed similar core gene counts, further highlighting the robustness of the core genome across different geographic populations (Supplementary Fig. 9). Furthermore, the non-US dataset displays a larger number of cloud genes, suggesting a higher diversity within its accessory genome.

Separate analyses of genomes from the United Kingdom (U.K.) ($n = 484$) and France ($n = 306$) were performed since they represented more than half of the non-U.S. genomes. Of 18 JI-groups defined in the

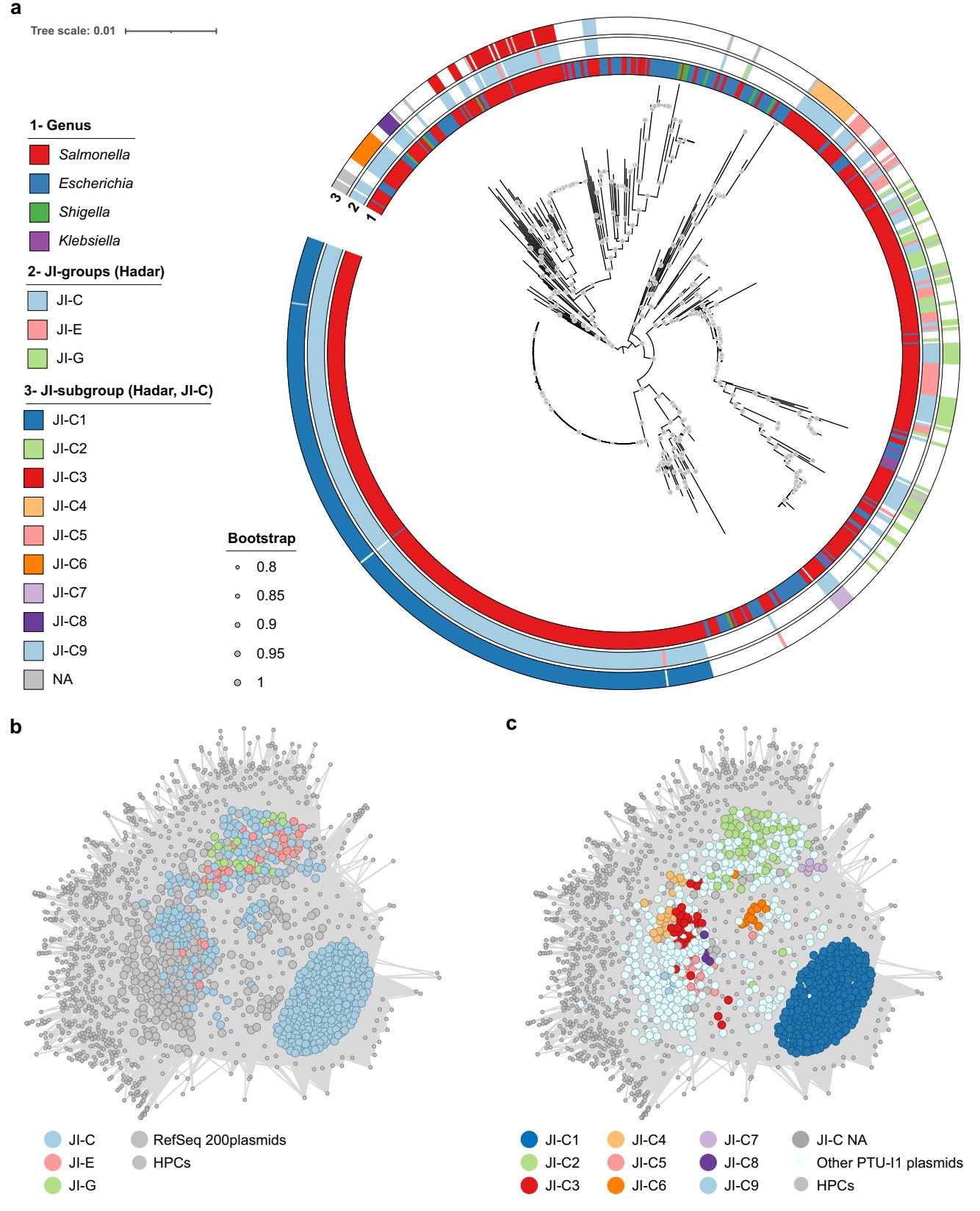

**a**

Tree scale: 0.01

**1- Genus**
- ■ *Salmonella*
- ■ *Escherichia*
- ■ *Shigella*
- ■ *Klebsiella*

**2- JI-groups (Hadar)**
- JI-C
- JI-E
- JI-G

**3- JI-subgroup (Hadar, JI-C)**
- JI-C1
- JI-C2
- JI-C3
- JI-C4
- JI-C5
- JI-C6
- JI-C7
- JI-C8
- JI-C9
- NA

**Bootstrap**
- 0.8
- 0.85
- 0.9
- 0.95
- 1

**b**

- JI-C
- JI-E
- JI-G
- RefSeq 200plasmids
- HPCs

**c**

- JI-C1
- JI-C2
- JI-C3
- JI-C4
- JI-C5
- JI-C6
- JI-C7
- JI-C8
- JI-C9
- JI-C NA
- Other PTU-I1 plasmids
- HPCs

U.S. dataset, the U.K. and France datasets shared only seven (170 genomes, 35%) and six (74 genomes, 24%) JI-groups, respectively. Seventeen U.K. JI-groups (228 genomes, 47%) and nine France JI-groups (162 genomes, 53%) were distinct from those isolated in the U.S (Supplementary Figs. 10 and 11, Supplementary Data 2). While no temporal shift was observed for pangenome groups from U.K. data

(Supplementary Fig. 10), a notable increase in genomes belonging to a novel group, JI-S, was observed in the France dataset, beginning in 2019 (Supplementary Fig. 11). JI-S genomes contain a prophage closely related to prophage 1, highlighting an intriguing parallel dynamic to the recent proliferation of prophage 1-containing groups JI-A and JI-C in the U.S. Thus, these analyses suggest Hadar pangenomic diversity is

**Fig. 7 | Core and protein analysis of plasmid taxonomic units (PTU)-I1 (IncI1) plasmids. a** Maximum likelihood (ML) core genome phylogenetic tree of 512 PTU-I1 plasmids from the Hadar dataset (see PTU-I1 plasmids of Jaccard Index (JI)-C, JI-E and JI-G in column "PTU" of Supplementary Data 1) and 259 PTU-I1 plasmids from RefSeq200 Enterobacterales hosts (Supplementary Data 5), generated using IQ-TREE v2[41]. The tree was midpoint rooted and visualized in iTol v6[41], UFBootstrap values > 80% are indicated by circles on the corresponding nodes, branch length scale represents substitutions per site. Ring 1 displays the plasmid host genus, Ring 2 displays the JI-group of plasmids found in Hadar, Ring 3 displays the JI-subgroup of the JI-C plasmids. **b** Proteome network of PTU-I1 plasmids colored by JI-group.

The proteins of the PTU-I1 plasmids were clustered at 80% identity and 80% coverage using AcCNET[43]. The larger nodes correspond to plasmids and are colored according to the JI-group of the Hadar plasmids (JI-C, JI-E, JI-G), or in grey if they belong to other Enterobacterales. The smaller nodes represent homologous protein clusters (HPCs) and are colored in grey. Both kinds of nodes are connected if a plasmid contains a member in the corresponding protein cluster. HPCs present in a single plasmid were removed. **c** Proteome network of PTU-I1 plasmids colored by JI-C-subgroup. The network was constructed as indicated in Fig. 7b. The JI-C Hadar plasmids are colored based on their JI-subgroup.

largely geographically defined, with potentially important genetic overlaps that will be further investigated.

## Discussion

Identifying the molecular mechanisms underlying shifts in bacterial populations is key to understanding the adaptive forces that drive evolution of human bacterial pathogens. Analysis of the Hadar pangenome confirmed known, and revealed unknown, epidemiological and microevolutionary dynamics. Before 2020, two distinct lineages separately dominated in commercial poultry (JI-B and JI-G) and backyard poultry environments (JI-D and JI-E). However, in 2020, an emergent lineage closely related to previously circulating strains became dominant, displacing the historical commercial poultry lineage. Around the same time, coinciding with a surge in backyard poultry ownership during the COVID-19 pandemic[16], this same emergent lineage became dominant among backyard poultry-associated human cases – confirming through high-resolution pangenomic analysis a link between two presumably separate industries. This finding underscores the interconnectedness of poultry industries and human health, highlighting the need for integrated and collaborative surveillance efforts.

Further, epidemiological and biological evidence suggest the presence of an uncharacterized phage in the emergent lineage may have contributed to its recent expansion. The role of this phage is critical, as it may represent a mechanism through which bacterial populations adapt and thrive in changing environments. Interestingly, a similar genetic shift underpinned by an emergent phage-containing lineage was seen in the French genomes analyzed here, suggesting this phenomenon is not restricted to the U.S. This observation raises important questions about the global nature of bacterial evolution and its implications for public health. The adaptive capacity of this prophage in Hadar, and specifically, the putative pathogenic role of the phage-encoded Zot-like protein, is still being evaluated in U.S. Hadar genomes.

These findings highlight opportunities that can be leveraged to mitigate further spread of this emergent strain. First, comparative plasmid analysis revealed a clonal subcluster of this lineage (JI-C1) that points to a reservoir upstream of backyard poultry suppliers and hatcheries, one that likely interfaces with commercial poultry. Understanding this reservoir is crucial for developing targeted interventions. The practices of backyard poultry hatcheries, such as drop-shipping and outsourcing to larger commercial hatcheries to meet demand[27,28], could explain this connection. These data can inform conversations between industry and government stakeholders, promoting collective action aimed at eliminating shared reservoirs affecting multiple industries. Second, functional analyses to determine the contribution of prophage 1 to avian gut colonization could inform intervention strategies in both commercial and backyard poultry settings; for example, by minimizing bacterial burden in birds, which is considered a control strategy to reduce risk of transmission to humans[29]. Such strategies are essential for protecting public health, particularly considering the evolving landscape of poultry ownership. Third, this analysis highlighted the importance of known MGE (e.g., PTU-I1 plasmids) and identified previously uncharacterized MGE (e.g.,

prophage 1) that can potentially be incorporated into source attribution models and molecular case definitions. Incorporating these findings into public health frameworks could enhance our understanding of transmission dynamics. For example, PTU-I1 (IncI1) plasmids could serve as a genetic marker that distinguishes backyard poultry-related strains from those transmitted via other sources. More accurate prediction of foodborne versus animal contact transmission pathways and refinement of outbreak and REP strain case definitions both contribute to timelier epidemiological traceback, and ultimately, a reduction in human illness[14].

More generally, this analysis enabled high-resolution genomic linking of human cases with potential sources, such as pet food and imported shrimp. This linkage raises suspicion of specific vehicles, which can refine supplemental interviews or traceback efforts when exposure information is limited and no transmission vehicles are otherwise suspected. Additionally, avenues were identified for investigation of ecological dynamics that underpin persistence of Hadar in different environments. For example, PTU-I1 (IncI1) and other large plasmids are associated with backyard poultry rather than commercial poultry environments. This distinction highlights the need for tailored surveillance strategies in different poultry sectors. Further, certain JI-groups (with unique MGE profiles) display a unique chicken association rather than the more common turkey signal.

Along with highlighting the previously unreported role of prophages in Hadar diversification and microevolution, this broad description of MGE in the U.S. Hadar population is foundational information for pathogen risk modeling, especially as it pertains to carriage of AMR. The presence of "risky" MGE related to AMR, virulence, or colonization capacity, can be proactively monitored through existing surveillance programs, and any emergent threats addressed before they become systematically disseminated, as has previously occurred with *Salmonella* serotypes Infantis[30] and Reading[31].

While the pangenomic approach employed here offers valuable insights, it also presents limitations. Exactly when and where this persisting REP strain arose was not determined; however, a molecular clock analysis is underway to explore the rapid rise and subsequent diversification of this lineage. Additionally, although source of human illnesses with unknown exposures, or those with multiple exposures (e.g., both commercial and backyard poultry), cannot be definitively determined using this approach, the findings from this study will be assessed within ongoing source attribution modeling to estimate the added value of inclusion of accessory genome content. Further, while efforts were made to obtain genomes representing diverse environments (wildlife, imported foods, commercial poultry production, backyard poultry environments, ill humans), several sources are underrepresented (e.g., live animals on farm) or absent (e.g., hatcheries), potentially missing pangenomic groups that are dominant in these spaces. Expanded analyses that include genomes from underrepresented sources, coupled with deeper investigation into the global diversity of Hadar, will fill important gaps in the pangenome landscape described here.

Unraveling pathogen epidemiology and microevolutionary dynamics is highly complex, and the plethora of available data is both an opportunity and a challenge. Leveraging existing genomic data, we

demonstrate the value of JI-based pangenomic analysis for delineating a highly clonal serotype and uncover actionable data to mitigate the spread of an emergent, and potentially more pathogenic, lineage of Hadar. We paint a pangenome landscape of this previously under-studied serotype, highlighting the importance of known and unknown MGE, and revealing surprising geographic patterns and dynamics. These findings will inform future risk and source attribution modeling, reducing public health burdens and mitigating impacts on implicated food and animal industries.

## Methods

This activity was reviewed by CDC and was conducted consistent with applicable federal law and policy (see e.g., 45 C.F.R. part 46, 21 C.F.R. part 56; 42 U.S.C. §241(d); 5 U.S.C. §552a; 44 U.S.C. §3501 et seq.).

### Data collection

A total of 3384 U.S. Hadar genomes were included in this analysis (Supplementary Data 1), collected between 1990 and 2023 (August 30th) from national surveillance systems and ad hoc sampling. Hadar genomes from ill humans with exposure information available were categorized as follows: "backyard poultry contact" – when contact was confirmed within seven days of illness onset (contact is defined as direct interaction with chickens, ducks, turkeys, geese, guinea fowl, or quail; direct contact with the environment where backyard poultry live and roam; consumption of eggs or meat obtained from backyard poultry; or residence with a household member who directly interacted with backyard poultry)[15], "turkey consumption" – where ground turkey was consumed within seven days of illness onset, and "unknown" – where exposure information was not available, or when neither backyard poultry contact nor turkey consumption was reported. Genomes from non-human sources were categorized according to the commodity from which they were sampled, for example, "commercial poultry" or "swine". "Other" was used to categorize samples from unknown food, animal, or environmental source types.

**National surveillance systems.** Salmonellosis is a nationally notifiable disease in the United States, and isolates obtained from patients are routinely submitted to public health laboratories (PHL) as part of the CDC's national enteric disease surveillance network, PulseNet USA[32]. Since 2019, PHL have performed whole genome sequencing (WGS) on all *Salmonella* isolates they receive and upload sequence data to a centralized national database for genetic analysis, including computed serotyping[32,33], and to the NCBI under the BioProject PRJNA230403. Additionally, public health departments routinely collect demographic information for all laboratory-confirmed cases of salmonellosis. For cases included in multistate outbreak investigations, public health officials conduct additional patient interviews, whenever possible, with supplementary standardized questionnaires to obtain further details about foods eaten and animal contact before illness onset[14]. ~5% of isolates detected by PHL also fall within the CDC arm of the National Antimicrobial Resistance Monitoring System (NARMS), a structured collection of enteric isolates from all 50 U.S. states used to monitor temporal trends in AMR (https://www.cdc.gov/narms/index.html). CDC NARMS has been routinely generating WGS data for this smaller subset of *Salmonella* isolates since 2016. WGS data for 2494 Hadar isolates collected between January 1st, 2016, and August 30th, 2023, were included in this analysis (Supplementary Data 1). For years prior to routine WGS (2005–2015), all Hadar isolates in PulseNet USA's national database with WGS data available were included (*n* = 55); these represent a small proportion of total isolates collected from this time period that were sequenced for various special interest projects.

The U.S. Food and Drug Administration (FDA) arm of NARMS routinely collects WGS data on *Salmonella* isolated from retail meats (chicken, ground turkey, ground beef, pork) purchased from U.S. grocery stores (https://www.fda.gov/animal-veterinary/national-antimicrobial-resistance-monitoring-system/about-narms). Sequencing data and sample source information are uploaded to the NCBI under the BioProject PRJNA292661. The following NCBI Pathogen Detection query (August 30th, 2023) identified 300 Hadar genomes (Supplementary Data 1) that were included in this analysis: https://www.ncbi.nlm.nih.gov/pathogens/isolates/#PRJNA292661%20AND%20Hadar.

The U.S. Department of Agriculture's Food Safety and Inspection Service (USDA-FSIS) routinely collects WGS data on *Salmonella* isolated from regulated food and animal products within U.S. food processing facilities (https://www.fsis.usda.gov/science-data/sampling-program/sampling-results-fsis-regulated-products). Sequencing data and sample source information are uploaded under NCBI BioProject PRJNA242847. Additionally, the USDA-FSIS arm of NARMS routinely collects WGS data from *Salmonella* isolated from the intestinal content of food animals at slaughter (https://www.fsis.usda.gov/science-data/national-antimicrobial-resistance-monitoring-system-narms) and data is uploaded under NCBI BioProject PRJNA292666. An August 30th, 2023 NCBI Pathogen Detection query identified 367 Hadar genomes from USDA-FSIS product sampling and 102 from NARMS sampling (Supplementary Data 1) for inclusion in this study: (https://www.ncbi.nlm.nih.gov/pathogens/isolates/#Hadar%20AND%20collected_by:USDA-FSIS).

**Ad hoc sampling systems.** To expand sample source type representation along the farm-to-fork continuum, Hadar genomes isolated from North America were included from ad hoc sampling systems. The FDA's Office of Regulatory Affairs (ORA), Center for Food Safety and Applied Nutrition (CFSAN), and Center for Veterinary Medicine (CVM) perform ad hoc WGS on human food (including imported) and animal food product samples and upload sequencing data to the Genome-Trakr project at NCBI (BioProject PRJNA186035). Twenty genomes (Supplementary Data 1) collected between 2003 and 2022 were selected and included in this analysis. An additional nine isolates representing all sequenced Hadar collected from sick animals as part of FDA-CVM's Veterinary Laboratory Investigation and Response Network (Vet-LIRN) AMR monitoring program were also included.

USDA's Animal and Plant Health Inspection Service (APHIS) provides ongoing animal disease surveillance and animal disease diagnostic services through the National Veterinary Services Laboratories (NVSL; https://www.aphis.usda.gov/labs/about-nvsl) and the National Animal Health Laboratory Network (NAHLN; https://www.aphis.usda.gov/labs/nahln). Thirty-two Hadar genomes (Supplementary Data 1) collected from chickens or turkeys from 2018 until 2023 as part of on-farm monitoring or for diagnostic purposes were included in this analysis. Three Hadar genomes previously sequenced and published by USDA's Agricultural Research Service (ARS)[34], and two Hadar genomes collected from wild ducks by the National Wildlife Health Center were also included (Supplementary Data 1). Additional Hadar genomes were available on NCBI, but sample source information availability (through NCBI or by request with submitter) was a requirement for inclusion in this analysis.

**Non-U.S. genomes.** A dataset of global non-U.S. Hadar genomes was generated from EnteroBase[35,36] for comparative analysis against the pangenome of the U.S. collection. All genomes with predicted serotype "Hadar" (EnteroBase employs SISTR1[37] and SeqSero2[38]) isolated in any country other than the U.S. were downloaded (n = 1145) (accessed December 21st, 2023) (Supplementary Data 2).

### Genomic analysis

Short reads with a base call quality score ≥ 28 and coverage ≥ 40x were assembled using shovill v.1.0.9 (https://github.com/tseemann/shovill) and resulting contigs with < 10% of the average genome coverage were excluded from the final assemblies. Serotype was confirmed using

SeqSero 2.0 v1.2.1[38], sequence type (ST) was determined using mlst (https://github.com/tseemann/mlst), core SNP cluster was obtained from NCBI Pathogen Detection's Isolate Browser (https://www.ncbi.nlm.nih.gov/pathogens/), and allele code was calculated from a 3,002 loci cgMLST schema, implemented in BioNumerics v7.6.3, described previously[3]. "Condensed allele code," which collapses allele codes to the third digit (e.g., *Salmonella* spp. allele codes SALM1.0 - 6771.1.1.30.1.21 and SALM1.0 - 6771.1.1.30.1.44 would be collapsed into SALM1.0 – 6771.1.1), was used to simplify representation of allele codes. Genomes of the same condensed allele code are expected to differ by less than -15 allele loci. Accessory (non-core) genome elements were detected using PanGraph v0.7.3 (see *Pangenome characterization*)[39] and characterized using PlasmidFinder[40] (updated July 17th, 2019; 90% identity, 60% gene coverage) for plasmid replicons, MOBscan[41] for conjugative relaxases (default parameters), CONJscan for detection of conjugative systems (implemented in MacSyFinder v2)[42,43], COPLA[44] for PTU designation[45], and Bakta v1.9.1 lightweight database[46] for gene annotation. AMR determinants, including acquired genes and chromosomal mutations, were detected using staramr v.0.4.0 (https://github.com/phac-nml/staramr?tab=readme-ov-file#mlsttsv), which employs the ResFinder database (updated July 30th, 2020; 90% identity, 50% gene coverage) and the *Salmonella* spp. PointFinder scheme[47]; predicted AMR was determined by staramr according to ResFinder and PointFinder results. Assignment of draft Illumina contigs to plasmids or chromosomes was performed using MOB suite v3.1.9[48].

Long-read sequencing was performed on 35 selected isolates representing each JI group (see *Jaccard Index calculation*), chosen strategically to maximize connectivity to other internal nodes and to best achieve JI-group representation. Eighteen Hadar isolates from people or food products were sequenced on the Oxford Nanopore GridION sequencing platform (Supplementary Data 3); reads were assembled using an in-house pipeline, as previously described[49]. Seventeen isolates collected from food or animal samples were sequenced using the 10-kb SMRTLink template preparation protocol (Pacific BioSciences, CA), as previously described[50]. Complete genomes were annotated by Bakta v1.9.1 lightweight database[46]. Long-read data are uploaded under BioSample numbers listed in Supplementary Data 3. An additional 18 previously published Hadar genomes[51] were also included in the analysis (Supplementary Data 3).

## Jaccard Index calculation

The exact JI was used as a measure of similarity between all genome pairs as previously reported[13]. Briefly, each genome assembly was converted into a set of *k*-mers. JI was calculated as the ratio of shared *k*-mers over the total number of different *k*-mers between the two sets (including shared *k*-mers, SNP *k*-mers differing by a single base pair, and indel *k*-mers differing between the datasets and excluding duplicated *k*-mers). BinDash v1.0[50] was employed to calculate JI, using parameters minhashtype = -1 (to compute the exact JI between highly similar genomes using the complete set of *k*-mers, rather than an estimated JI based on a subset of *k*-mers) and *k*-mer length (k)=21 (as previously defined as optimum in ref. 13) (https://github.com/PenilCelis/Salmonella_Typhi_JINA).

## Network visualization and community detection

The adjacency matrix of pairwise genome similarities generated by BinDash was used to construct an undirected network. Gephi v10[52] was employed to visualize the network, using the ForceAtlas2 algorithm for the layout. To define the final components for study, referred to as JI-groups, a range of JI thresholds was assessed to balance excessive fragmentation at higher values and over-clustering at lower values. Network sparsification was optimized according to transitivity and density, as previously described[13]. Transitivity plateaued between JI 0.986 and 0.990, indicating consistent internal cluster connectivity

within this range. The final JI threshold was set in the middle of this range, at 0.988, balancing the density of communities and the number of singletons.

The Louvain method, implemented in Gephi, was used to define the JI-groups by using resolution 1.5. Once the main JI groups are defined (containing a minimum of five genomes), they can be further dissected into several subgroups within the network using a more stringent JI and the same community detection algorithm[13]. The nodes of the network, representing genomes, were colored according to metadata and genetic determinants of interest. Edges between nodes were included whenever the corresponding JI value met or exceeded the user-defined threshold. Network figures were generated using the igraph package in R.

## Pangenome characterization

PanGraph v0.7.3[39] identifies blocks of homologous sequence and was used to detect indels specific to each JI-group. PanGraph was run on all genomes using parameters 'α = 20' and 'β = 20'. The parameter α controls the cost of splitting a block into smaller units, where a value of 20 was chosen to minimize excessive fragmentation of the graph. The parameter β controls the diversity cost and was set to 20, establishing a sequence diversity threshold of 20%. Only homologous sequences (pancontigs), larger than 250 bp, present in ≥85% of the members of each JI-group and not present in all JI-groups, were retained as "core" pancontigs (Supplementary Data 4). Core pancontigs for each JI-group were mapped with BLASTn (BLAST+ v2.15.0) against a reference genome from their respective JI-group (Supplementary Data 1; sequenced by long-read technology, when available) to order the pancontigs and detect the regions they form. For instance, a prophage might be composed of several pancontigs, and scaffolding those contigs against a reference genome helped reconstruct and identify that element as an indel. The term "prophage" was used to refer to chromosomally-integrated regions that contained at least five phage-related genes according to PhageScope v1.2.1[53] or PHASTEST v3.0[54].

Roary v3.13[55] was used for pangenome comparison between U.S. and non-U.S. datasets. Following the recommendations for Roary, Prokka v1.14.6[56] was used for gene prediction of the assembled genomes, and the resulting GFF3 files were used as input, with a threshold of 80% protein identity and coverage. Pangenome gene categories were defined as: core genes (shared by 80–100% of the genomes); shell genes (15–79%); and cloud genes (0–14%). Heaps' law was used to evaluate pangenome openness and closeness, using the script available at (https://github.com/SethCommichaux/Heap_Law_for_Roary).

## Phylogenetic analysis

cgMLST-based phylogenetic trees were generated using BioNumerics v7.6.3[3]. Snippy v4.6 (https://github.com/tseemann/snippy) was used to detect core genome single nucleotide polymorphisms (cg-SNPs) in three datasets: JI-C chromosomes, JI-C PTU-I1 (IncI1) plasmids, and PTU-I1 (IncI1) plasmids from JI-E, JI-G, and other Enterobacteria (RefSeq200) (Supplementary Data 5). In all cases, the PTU-I1-containing genome SAL-20-VL-OH-OSU-0008 was used as a reference. Alignments generated with Snippy were used to construct maximum-likelihood (ML) phylogenetic trees based on cg-SNPs by using IQ-TREE v2.3.3 with the ultra-fast bootstrap option[57]. All trees generated in this study were rooted at midpoint and visualized with iTol v6[58]. To complement cg-SNP analysis of PTU-I1 (IncI1) plasmids, AcCNET[59] was used to build proteome networks and assess relatedness of plasmids at the protein level; proteins were clustered if they shared greater than 80% identity and 80% coverage.

## Statistical analysis

Statistical analyses were performed using genomes collected through NARMS (CDC, FDA, FSIS), PulseNet (CDC), and FSIS national

surveillance systems from years 2016 through 2023, in line with the introduction of routine sequencing for NARMS, PulseNet, and FSIS surveillance isolates. Corrected Cramer's $V$[60] was used to measure strength of associations between JI-groups and epidemiological (e.g., year, state, source of isolation, patient demographics) and genomic (e.g., allele code, plasmids, AMR determinants) categorical variables of interest (Supplemental Table 1). Chi-squared tests of independence were used to test associations between specific epidemiological and genomic variables (Bonferroni adjusted significance value for multiple comparisons: $p < 0.005$), and odds ratios (OR) (95% CI) were used to quantify the strength and direction of those significant associations. For statistical tests involving a specific JI-group, the comparison group was always "all other JI-groups". All tests were calculated using the stats subpackage of SciPy v1.14.1 implemented in Python v3.11.7 (https://docs.scipy.org/doc/scipy/reference/stats.html). JI-groups with less than 20 genomes were not analyzed for statistical associations. Only NARMS surveillance data collected by CDC, FDA, and FSIS (cecal sampling) were used to assess shifts in pangenome group abundance over time, as the isolates in the NARMS dataset were systematically collected and were more robust against large outbreaks and changes to regulatory testing practices than were the surveillance isolates from the PulseNet and FSIS product sampling datasets. Map figures were visualized in R v4.4.0 using the ggplot2 v3.5.2, dplyr v1.1.4, tidyr v1.3.1, gridExtra v2.3, scatterpie v0.2.5, RcolorBrewer v1.1-3, usmap v0.8.0, and sf v1.0-21 packages.

### Reporting summary

Further information on research design is available in the Nature Portfolio Reporting Summary linked to this article.

## Data availability

The authors declare that all genomic data supporting the findings of this study are publicly available through NCBI and Enterobase using accession numbers listed within the paper and its supplementary information files available through Figshare. Epidemiological data is available within supplementary information files. Some human patient information collected as part of routine public health surveillance or through supplementary standardized questionnaires are not publicly available due to data privacy laws; deidentified data are available on request by contacting pulsenet@cdc.gov, per data sharing policies. Source data are provided with this paper.

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

## Acknowledgements

The authors would like to acknowledge state and local public health departments and laboratories for isolation and sequencing of Hadar genomes included in this analysis. The authors thank FDA colleagues Olgica Ceric, Beilei Ge, Claudine Kabera, as well as University of Minnesota Professor Timothy Johnson, for their valuable expertise. This work was supported by the Centers for Disease Control and Prevention (Contract No. 75D30123P18303 to FdlC). This work was also supported by the Spanish Ministry of Science and Innovation MCIN/AEI/10.13039/501100011033 (PID2020-117923GB-I00 to FdlC and MPGB). The views expressed in this article are those of the authors and do not necessarily reflect the official policy of the Agencies within the U.S. Department of Health and Human Services (CDC, FDA) and the U.S. Department of Agriculture (FSIS, APHIS), or the U.S. Government. Mention of trade names or commercial products in this publication is solely for the purpose of providing specific information and does not imply recommendation or endorsement by the U.S. Department of Health and Human Services or the U.S. Department of Agriculture.

## Author contributions

K.A.T., A.P.C., H.E.W., G.S.S., M.K.S., K.B., M.P.G.B. and F.dlC.–Conceptualization. K.A.T., A.P.C., H.E.W., G.S.S., Z.E., M.L., J.Y.K., M.S., C.L., B.H., B.R.M.S., D.M., S.M., K.M., J.H., J.M.W., J.M.B. and K.B.–Data collection. K.A.T., A.P.C., H.E.W., G.S.S., M.S., C.L., B.H., B.R.M.S., K.B. and U.D.–Data curation. K.A.T., A.P.C., H.E.W., M.K.S.,

S.R.S., M.P.G.B. and Fdl.C.–Methodology. K.A.T., A.P.C., H.E.W., M.K.S., S.R.S., and M.P.G.B.–Analysis. K.A.T., A.P.C., H.E.W., M.K.S., S.R.S., M.P.G.B. and Fdl.C.–Visualization. K.A.T., A.P.C., H.E.W., G.S.S., K.B., M.P.G.B. and Fdl.C.– Writing - original draft. K.A.T., A.P.C., H.E.W., G.S.S., Z.E., M.L., J.Y.K., M.S., G.T., C.L., B.H., B.R.M.S., M.K.S., D.M., S.M., K.M., J.H., J.M.W., C.S., J.M.B., S.S., K.B., J.P.F., U.D., S.R.S., M.P.G.B. and Fdl.C.–Writing - review and editing. All authors read and approved the final manuscript.

## Competing interests

The authors declare no competing interests.

## Additional information

[1]Division of Foodborne, Waterborne and Environmental Diseases, Centers for Disease Control and Prevention, Atlanta, USA. [2]Instituto de Biomedicina y Biotecnología de Cantabria, Consejo Superior de Investigaciones Científicas-Universidad de Cantabria, Santander, Spain. [3]Oak Ridge Institute for Science and Education, Oak Ridge, USA. [4]ASRT, Inc, Smyrna, USA. [5]Food Safety and Inspection Service, United States Department of Agriculture, Washington D.C, USA. [6]Center for Veterinary Medicine, United States Food and Drug Administration, Laurel, USA. [7]National Animal Health Laboratory Network, Animal and Plant Health Inspection Services, U.S. Department of Agriculture, Ames, USA. [8]National Veterinary Services Laboratories, Animal and Plant Health Inspection Service, U.S. Department of Agriculture, Ames, USA. [9]Colorado Department of Public Health and Environment, Glendale, USA. [10]Montana Public Health Laboratory, Helena, USA. [11]Utah Public Health Laboratory, Taylorsville, USA. [12]These authors contributed equally: Kaitlin A. Tagg, Arancha Peñil-Celis. ✉e-mail: arancha.penil@unican.es; hwebb@cdc.gov

