## [Peer Review File · Nature Communications]

Pangenome dynamics and population structure of the zoonotic pathogen *Salmonella enterica* serotype Hadar

Corresponding Author: Dr Hattie Webb

Version 0:

Reviewer comments:

Reviewer #1

(Remarks to the Author)

Thank you for the opportunity to review the manuscript entitled “The *Salmonella enterica* serotype Hadar pangenome: Population structure and dynamics of a zoonotic pathogen” Below, I provide a brief summary of the study, followed by my overall assessment, questions for the authors, and suggestions for improvement.

The manuscript presents a comprehensive pangenomic analysis of the *Salmonella enterica* serotype Hadar, a zoonotic pathogen of increasing importance in the United States. Using a k-mer-based approach grounded in the Jaccard index, the authors investigate over 3,300 genomes from human, animal, and environmental isolates collected over three decades. The study reveals notable shifts in the population structure of *Salmonella* Hadar, including the emergence of a dominant lineage associated with prophage 1 and its spread in both backyard and commercial poultry environments. The authors demonstrate the utility of pangenome analysis for epidemiological inference, source attribution, and surveillance of recurrent lineages. The article is very well written, with a logical structure, clear language, and a solid methodological foundation. The topic is highly relevant to public health and is contextualized accurately. The methodology is modern and well executed, with appropriate use of computational and bioinformatic tools. The results are detailed and thoroughly discussed, highlighting the genomic and epidemiological implications of the emergence of new lineages. The manuscript also addresses its main limitations and proposes promising future directions.

Questions for the authors

Are there any plans to conduct experimental functional assays to assess the role of prophage 1 (in particular, the Zot-like gene) in the virulence or adaptation of *Salmonella* Hadar?

Considering the importance of PTU-I1 plasmids, is there any evidence of recent horizontal transfer between different environments (e.g., between backyard chickens and commercial production)?

Suggestions for improvement

Standardize the nomenclature of mobile genetic elements (MGEs): In some sections, there are inconsistencies in how plasmids and prophages are presented, such as “PTU-I1 (Incl1)” versus “Incl1 plasmid of PTU-I1.” It is recommended to choose a consistent format throughout the text and define all acronyms at first mention (e.g., PTU, ICE, Tn, AR).

Improve the flow of transitions between sections: Some passages (particularly in the Results and Discussion) shift quickly from technical descriptions to epidemiological implications. Adding transition sentences would help maintain argumentative coherence.

Include a geographic map of the isolates: Given the temporal and regional analyses (with emphasis on states such as California), a spatial visualization could help illustrate the spread and emergence of lineages.

Incorporate key supplementary information into the main text: Some critical details (e.g., the definition of JI groups, differential mobile elements) are found only in the supplementary tables. A summarized version could be included in a table within the main body of the article.

Reviewer #2

(Remarks to the Author)

The present paper entitled “The *Salmonella enterica* serotype Hadar pangenome: Population structure and dynamics of a zoonotic pathogen” by Kaitlin A. Tagg and colleagues convincingly presents an important body of work surrounding a recent

microevolutionary event in *S. Hadar*, an important pathogen occurring in poultry in the US, and a considerable cause of recent human illnesses. Their work suggests that a novel lineage emerged from extant populations circulating in commercial poultry, with subsequent dissemination into backyard poultry environments. The present work is highly relevant and offers methodological advances, demonstrating the value of JI-based pangenomic analysis for delineating a highly clonal serotype, and offers support for the incorporation of the accessory genome (in addition to the core genome via cgMLST) for differentiating strains transmitted via different pathways. These aspects are likely to be generally applicable to other serovars or even taxa of bacterial pathogens and as such, hold substantial value to the scientific community. The work also includes actionable considerations regarding how the present findings in *S. Hadar* can inform management strategies and potentially mitigate the spread of this emergent strain.

I have a number of relatively minor points, that the authors should address to make this manuscript suitable for publication in Nature Communications:

1. In section "Genome analysis" (page 9) Bakta is used for accessory genome gene prediction and long-read assembly genome annotation. In a later section, ("Pangenome characterization", page 11), Prokka is used. Were both tools used, and if so, why and for which purpose (i.e. were results from different tools used for different downstream analyses)? Why wasn't Bakta (which essentially is Prokka's successor) used for all analyses for consistency and comparability? Also, which DB version was used?
2. Page 12, "Corrected Cramer's V was used to measure strength of associations between all categorical variables": It is not entirely clear which categorical variables were included in association testing. Are they only the ones mentioned in "Data collection" (i.e. exposure information and source type)? On page 4 it says "epidemiological and source information", other sections of the Results mention that other variables such as geographic location, processing facilities and patient demographics (age, sex, site of infection, hospitalization) were also tested for associations. From the methods on Page 5 and Supplementary Table S1 it is unclear which variables were used for association testing. I suggest both briefly summarizing the tested epidemiological and genomic associations in the Methods as well as adding a supplementary table, listing all tests and their respective test metrics similar to what was done for Supplementary Table S4.
3. Page 6, "Supplementary standardized questionnaires" should read "supplementary standardized questionnaires" (lower case).
4. Page 8, last sentence: reference 18 is used here to reference cgMLST. However, this paper only describes the implementation of WGS-based methods by PulseNet and mentions the schemes used, but does not mention which tool is used to determine cgMLST. Reference 3 (Leeper et al. 2023; <https://doi.org/10.3389/fmicb.2023.1254777>), by one of the co-authors, seems more appropriate here, as it mentions the tool and approaches used, which should also be briefly mentioned in the Methods.
5. Page 16: "In contrast, groups JI-D and JI-E were almost always from ill humans (rather than animal or meat samples) (Fig. 3e), often with reported contact with backyard poultry (Supplementary Table S1)": I checked Supplementary Table S1, and from the 231 isolate belonging to JI-D and JI-E all are SourceTypeCondensed "human clinical", so I suggest to remove "almost always" and "(rather than animal or meat samples)" in the first half of this sentence. For the second half of the sentence, of the 231 isolates, 103 (44.6%) were SourceType/SampleType "backyard poultry contact", which may or may not be considered "often". I suggest to include these percentages here for clarity/transparency.
6. Page 17: "In contrast, JI-A3 was almost exclusively genomes..." to "In contrast, JI-A3 was almost exclusively comprised of genomes..."
7. Page 22, last paragraph: "a molecular clock analysis is underway to explore the rapid rise and subsequent diversification of this lineage". I understand that there is a limit to what can be included in the study and the authors are certainly entitled to not include this analysis here. However, given that no additional original data would have to be created for this analysis (I assume) and since this analysis would add substantial value to the present work, I would hope that the authors reconsider including this additional work. If the authors have compelling reason to not include these analyses in the present study, I ask them to briefly outline the reasons for doing so.
8. Figure 1: coloring of individual JI groups is hard to distinguish, I suggest choosing higher contrast colors.
9. Figure 2: Increase overall font size and color legend.
10. Figure 3: Colors not distinguishable, see comment on Fig 1. Color legends need to be larger, text is not readable. Consistent formatting of panels ("a" in the caption vs. "A" in the figure).
11. Figure 4: This is an excellent overview figure depicting isolates belonging to different JI clusters and isolation sources over time. It also highlights the potential for biases introduced by routine and opportunistic/ad hoc sampling approaches (also in the context of the COVID pandemic, which is briefly mentioned in the introduction, page 4), but I believe the authors have sufficiently addressed these limitations in the Discussion. If any other clear biases exist and are known to the authors, they could be added to this section in the Discussion, which at this point, is of a more general nature.
12. Figure 5: The colors of the individual rings have very low contrast to each other, which makes it hard to read. The color legend and labels need to be increased in size. "21/3384 could not be processed through BioNumerics": Why is that? I could not find any information on this.
13. Figure 6: Here the coloring and sizing of labels is better, I suggest adopting this for the other figures. Fix panel label formatting ("a" vs. "A").

Version 1:

Reviewer comments:

Reviewer #1

(Remarks to the Author)

Thank you for addressing my comments, edits, and suggestions, and for providing a revised and improved version of the manuscript. I have no further recommendations.

Reviewer #2

(Remarks to the Author)

I thank the authors for thoroughly addressing my comments and feedback. I have no further objections to the publication of this article. Since an additional Figure has been added (Figure 5), the column header of column AA in Supplementary Data 1 should be changed from "Figure 5" to "Figure 6" (cf. comment number 12 of my review).

RESPONSE TO REVIEWER COMMENTS

Reviewer #1 (Remarks to the Author):

Questions for the authors

Are there any plans to conduct experimental functional assays to assess the role of prophage 1 (in particular, the Zot-like gene) in the virulence or adaptation of Salmonella Hadar?

Yes! As we don't have the laboratory capacity in-house to conduct such experiments, we are partnering with another federal agency to complete this work. To start, naturally occurring +/- prophage Hadar strains and knock-out mutants are to be analyzed and compared by Biolog Phenotype Microarrays (which may identify microbial systems and phenotypes to explore deeper) as well as cell culture assays for attachment, invasion and survival. This work is anticipated to start before the end of 2025.

Considering the importance of PTU-II plasmids, is there any evidence of recent horizontal transfer between different environments (e.g., between backyard chickens and commercial production)?

We thank the reviewer for this important question. Most genomes in our dataset were sequenced using Illumina short reads, and therefore the plasmids are not always closed. However, for a subset of isolates we also have Nanopore long-read data, which allowed us to obtain fully closed plasmid sequences. While acknowledging these limitations, the available data suggest recent horizontal transfer of PTU-II plasmids between different genomic backgrounds and environments.

PTU-II plasmids are present in isolates grouped in different allele codes and JI-groups (JI-C, JI-E, and JI-G). As shown in the tree of Figure 7 (revised draft), PTU-II plasmids exhibit differences in their core genome, with instances of plasmids belonging to the same JI group placed in different phylogenetic clades of the tree. Notably, there are clades containing closely related plasmids (ANI values > 99.9%), which are carried by isolates of different JI-groups, suggesting transfer of the plasmid between different genomic backgrounds (highlighted with a red arrow in the figure below). Moreover, a subset of these isolates from different JI-groups were obtained from different environments (e.g. practically identical plasmids (ANI=99.9904%) present in an isolate of the JI-G group isolated from commercial poultry and in an isolate of the JI-E group isolated from a human clinical sample with reported backyard poultry contact). This pattern indicates that PTU-II plasmids may have been horizontally transferred between strains circulating in distinct environments.

Finally, we also detected cross-species similarities: for example, a PTU-II plasmid in *E. coli* was >99.9% identical to that found in JI-C1 (highlighted with a green arrow in the figure below), and another *E. coli* plasmid showed high similarity to PTU-II plasmids from JI-C3 and JI-E (highlighted with a blue arrow in the figure below). All these findings taken together indicate that

PTU-II plasmids are not restricted to *Salmonella* lineages but circulate across different bacterial species as well.

We have modified the text to incorporate the referee's suggestion:

“PTU-II (IncII) plasmids from all three JI-groups were surprisingly diverse in their core and proteome. They did not form phylogenetic clades defined by the JI-group of the host, instead plasmids from the same JI-group were clustered in different clades (Fig. 7a). Notably, nearly identical plasmids were found in isolates recovered from different environments; for example, in a JI-G isolate from commercial poultry (FSIS11705123) and in a JI-E isolate from a human clinical case with reported backyard poultry contact (PNUSAS013855). PTU-II (IncII) plasmids also intermingled phylogenetically with those from other Enterobacteriaceae species (Fig. 7a and 7b, Supplementary Data 4); for instance, one E. coli plasmid was > 99.9% identical to a JI-C1 plasmid (Fig. 7a). These findings support the notion that PTU-II (IncII) plasmids move horizontally between strains circulating in different environments and across different bacterial species.”

Suggestions for improvement

Standardize the nomenclature of mobile genetic elements (MGEs): In some sections, there are inconsistencies in how plasmids and prophages are presented, such as “PTU-II (IncII)” versus “IncII plasmid of PTU-II.” It is recommended to choose a consistent format throughout the text and define all acronyms at first mention (e.g., PTU, ICE, Tn, AR).

Edits have been made to standardize in the main text and Supplementary Information.

Improve the flow of transitions between sections: Some passages (particularly in the Results and Discussion) shift quickly from technical descriptions to epidemiological implications. Adding transition sentences would help maintain argumentative coherence.

Thank you for this feedback. Transitions have been thoughtfully reviewed and adjusted.

Include a geographic map of the isolates: Given the temporal and regional analyses (with emphasis on states such as California), a spatial visualization could help illustrate the spread and emergence of lineages.

We agree this would be helpful and have added a new figure (now Figure 5) to the manuscript, and it enable us to expand the geographic discussion in the “*Genetic and epidemiological differences between pangenome groups*” section of the Results and refer to this new figure in some of the preexisting results.

Incorporate key supplementary information into the main text: Some critical details (e.g., the definition of JI groups, differential mobile elements) are found only in the supplementary tables. A summarized version could be included in a table within the main body of the article.

We appreciate this feedback. The table included in Figure 1 has now been updated to include the dominant SNP cluster and PTU profile of each JI-group. The differential mobile elements for each JI-group are shown in Figure 2.

Reviewer #2 (Remarks to the Author):

I have a number of relatively minor points, that the authors should address to make this manuscript suitable for publication in Nature Communications:

1. In section “Genome analysis” (page 9) Bakta is used for accessory genome gene prediction and long-read assembly genome annotation. In a later section, (“Pangenome characterization”, page 11), Prokka is used. Were both tools used, and if so, why and for which purpose (i.e. were results from different tools used for different downstream analyses?)?

Why wasn’t Bakta (which essentially is Prokka’s successor) used for all analyses for consistency and comparability? Also, which DB version was used?

Both annotation tools were indeed used, but for different purposes. Bakta, regarded as the successor of Prokka, was employed as our main annotation tool and applied to complete genomes, as it provides more up-to-date and accurate protein annotations. This was particularly important for accessory genome characterization (plasmids, regions of interest, etc.). In the Methods section we have included the information on the version and database (“Bakta v1.9.1 using lightweight database”).

In contrast, Prokka was used for the pangenome calculation step with Roary, in the comparison of US vs non-US genomes. Roary requires GFF3 files as input. While both Bakta and Prokka generate GFF3 files, their formats are not identical. Since Roary is optimized for the GFF3 files produced by Prokka and the official Roary documentation (<https://sanger-pathogens.github.io/Roary/>) explicitly recommends using Prokka, we followed this guideline to

ensure compatibility and consistency. We clarified this point and added the version of Prokka in the Methods section.

Importantly, Prokka results were not mixed with Bakta annotations; Prokka was used exclusively as input for Roary to maintain methodological consistency.

2. Page 12, “Corrected Cramer’s V was used to measure strength of associations between all categorical variables”: It is not entirely clear which categorical variables were included in association testing. Are they only the ones mentioned in “Data collection” (i.e. exposure information and source type)?

On page 4 it says “epidemiological and source information”, other sections of the Results mention that other variables such as geographic location, processing facilities and patient demographics (age, sex, site of infection, hospitalization) were also tested for associations. From the methods on Page 5 and Supplementary Table S1 it is unclear which variables were used for association testing. I suggest both briefly summarizing the tested epidemiological and genomic associations in the Methods as well as adding a supplementary table, listing all tests and their respective test metrics similar to what was done for Supplementary Table S4.

We agree the tested associations need to be clarified. A supplemental table has been added listing the categorical variables that were tested using corrected Cramer’s V . Additionally, text has been added in the methods section giving examples of the categorical variables that were tested with this method. It now reads:

Corrected Cramer’s V [46] was used to measure strength of associations between JI-groups and epidemiological (e.g. year, state, source of isolation, patient demographics) and genomic (e.g. allele code, plasmids, AR determinants) categorical variables of interest (Supplementary Table 1).

3. Page 6, “Supplementary standardized questionnaires” should read “supplementary standardized questionnaires” (lower case).

This edit has been made.

4. Page 8, last sentence: reference 18 is used here to reference cgMLST. However, this paper only describes the implementation of WGS-based methods by PulseNet and mentions the schemes used, but does not mention which tool is used to determine cgMLST. Reference 3 (Leeper et al. 2023; <https://doi.org/10.3389/fmicb.2023.1254777>), by one of the co-authors, seems more appropriate here, as it mentions the tool and approaches used, which should also be briefly mentioned in the Methods.

We thank the reviewer for pointing this out. This citation has been updated, and an edit has been made to the sentence in the methods. It now reads as follows:

...allele code was calculated from a 3,002 loci cgMLST schema, implemented in BioNumerics v7.6.3, described previously [3].

5. Page 16: “In contrast, groups JI-D and JI-E were almost always from ill humans (rather than animal or meat samples) (Fig. 3e), often with reported contact with backyard poultry (Supplementary Table S1)”: I checked Supplementary Table S1, and from the 231 isolate belonging to JI-D and JI-E all are SourceTypeCondensed “human clinical”, so I suggest to remove “almost always” and “(rather than animal or meat samples)” in the first half of this sentence. For the second half of the sentence, of the 231 isolates, 103 (44.6%) were SourceType/SampleType “backyard poultry contact”, which may or may not be considered “often”. I suggest to include these percentages here for clarity/transparency.

We agree that percentages are needed for clarity and transparency; the sentence has now been updated to address this (see below). Of note, JI-E is comprised entirely of human clinical Hadar, but JI-D has six genomes from non-human sources.

In contrast, groups JI-D and JI-E were almost exclusively from ill humans (n=6/191 JI-D genomes are from non-human sources) (Fig. 3f), with upwards of 40% of cases reporting contact with backyard poultry (n=79/191 JI-D genomes; n=24/40 JI-E genomes, Supplementary Data 1).

6. Page 17: “In contrast, JI-A3 was almost exclusively genomes...” to “In contrast, JI-A3 was almost exclusively comprised of genomes...”

Accepted suggestion.

7. Page 22, last paragraph: “a molecular clock analysis is underway to explore the rapid rise and subsequent diversification of this lineage”. I understand that there is a limit to what can be included in the study and the authors are certainly entitled to not include this analysis here. However, given that no additional original data would have to be created for this analysis (I assume) and since this analysis would add substantial value to the present work, I would hope that the authors reconsider including this additional work. If the authors have compelling reason to not include these analyses in the present study, I ask them to briefly outline the reasons for doing so.

Thank you for the helpful feedback. We definitely understand the value of a molecular clock analysis.

There are several questions of interest that could be addressed with molecular clock analyses. Firstly, when and where the REPTDK01 strain first emerged, and when JI-A/JI-C shared a most recent common ancestor with JI-B (these groups fall within the same NCBI SNP cluster and thus are related). This would require a diverse subset of strains representing circulating populations of Hadar strains over time. The second question is regarding if and when subclades of the REPTDK01 strain emerged separately in the distinct industries of backyard poultry versus commercial poultry. This requires a focused analysis of REPTDK01 strains.

For the former, we generated a core SNP tree using a subset of genomes that were systematically collected over time from humans, retail meats, and animals (NARMS surveillance sampling). This subset included the REPTDK01 strain. We tested the “clocklikeness” signal of this tree using Tempest, which is the first step in determining whether a molecular clock is appropriate for analyzing the population of genomes. Unfortunately, this diverse set of genomes did not meet the assumptions of a molecular clock, in that branch lengths did not correlate well with isolation dates ($R^2 = 0.032$). Thus, this subset of diverse, representative genomes is not a good candidate for performing further molecular clock analysis. We have included the data from Tempest below.

For the latter, colleagues within CDC utilized a

subset of our dataset to generate a focused molecular clock analysis of this particular strain, REPTDK01. This manuscript serves as a deep dive into the recent divergence of REPTDK01 and is close to being submitted to a peer-reviewed journal. Since the manuscript being reviewed here explores the broader Hadar population in the US and globally, and thoroughly describes Hadar pangenomic diversity, we thought it best to keep the focused REPTDK01 molecular clock analysis separated into a complementary paper.

8. Figure 1: coloring of individual JI groups is hard to distinguish, I suggest choosing higher contrast colors.

Thank you for the suggestion. This figure has been regenerated using higher contrast colors. These colors have been carried through others figures in the manuscript for consistency.

9. Figure 2: Increase overall font size and color legend.

This figure has been updated to increase font size and color legend.

10. Figure 3: Colors not distinguishable, see comment on Fig 1. Color legends need to be larger, text is not readable. Consistent formatting of panels (“a”) in the caption vs. “A” in the figure).

We have increased the legend text size and updated the colors to be more distinguishable. The figure caption has also been updated to be consistent with the figure.

11. Figure 4: This is an excellent overview figure depicting isolates belonging to different JI clusters and isolation sources over time. It also highlights the potential for biases introduced by routine and opportunistic/ad hoc sampling approaches (also in the context of the COVID pandemic, which is briefly mentioned in the introduction, page 4), but I believe the authors have sufficiently addressed these limitations in the Discussion. If any other clear biases exist and are known to the authors, they could be added to this section in the Discussion, which at this point, is of a more general nature.

Thank you for the feedback. We believe the major biases in our national and ad hoc surveillance sampling have been mentioned in the text. Additionally, we have updated Figure 4 to reflect the JI-group color changes suggested by the reviewer.

12. Figure 5: The colors of the individual rings have very low contrast to each other, which makes it hard to read. The color legend and labels need to be increased in size. “21/3384 could not be processed through BioNumerics”: Why is that? I could not find any information on this.

We agree the contrast on this figure can be improved. We have altered the colors to be easier to read and increased the font size and resolution of the image. This tree was generated within BioNumerics, and in some cases, the whole genome sequencing data cannot be linked to the isolate record. Unfortunately, because we only have access to the graphical user interface, we have no ability to link data on the back end. The 21/3384 genomes that could not be processed through BioNumerics are noted in Supplementary Data 1.

13. Figure 6: Here the coloring and sizing of labels is better, I suggest adopting this for the other figures. Fix panel label formatting (“a”) vs. “A”).

Formatting labels addressed in panel label.

REVIEWERS' COMMENTS

Reviewer #1 (Remarks to the Author):

Thank you for addressing my comments, edits, and suggestions, and for providing a revised and improved version of the manuscript. I have no further recommendations.

Reviewer #2 (Remarks to the Author):

I thank the authors for thoroughly addressing my comments and feedback. I have no further objections to the publication of this article. Since an additional Figure has been added (Figure 5), the column header of column AA in Supplementary Data 1 should be changed from "Figure 5" to "Figure 6" (cf. comment number 12 of my review).

Thank you for catching this detail. Column header of column AA in Supplementary Data 1 has been updated to read Figure 6. We have also added Figure 4 and Figure 5.